# Diagnosis of Classic Homocystinuria in Two Boys Presenting with Acute Cerebral Venous Thrombosis and Neurologic Dysfunction after Normal Newborn Screening

**DOI:** 10.3390/ijns7030048

**Published:** 2021-07-23

**Authors:** Alexander Asamoah, Sainan Wei, Kelly E. Jackson, Joseph H. Hersh, Harvey Levy

**Affiliations:** 1Norton Children’s Medical Group, University of Louisville School of Medicine, Louisville, KY 40202, USA; kelly.jackson@louisville.edu (K.E.J.); joseph.hersh@louisville.edu (J.H.H.); 2Department of Pathology and Laboratory Medicine, University of Kentucky, Lexington, KY 40536, USA; sainan.wei@ky.gov or; 3Division of Laboratory Services, Kentucky Department for Public Health, Frankfort, KY 40601, USA; 4Division of Genetics and Genomics, Boston Children’s Hospital, Boston, MA 02115, USA; harvey.levy@childrens.harvard.edu; 5Department of Pediatrics, Harvard Medical School, Boston, MA 02115, USA

**Keywords:** homocystinuria, HCU, newborn screening, homocysteine, CBS gene, next generation sequencing, NGS, dried blood spot, DBS

## Abstract

Homocystinuria, caused by cystathionine β-synthase deficiency, is a rare inherited disorder involving metabolism of methionine. Impaired synthesis of cystathionine leads to accumulation of homocysteine that affects several organ systems leading to abnormalities in the skeletal, cardiovascular, ophthalmic and central nervous systems. We report a 14-month-old and a 7-year-old boy who presented with neurologic dysfunction and were found to have cerebral venous sinus thromboses on brain magnetic resonance imaging (MRI)/magnetic resonance venogram (MRV) and metabolic and hypercoagulable work-up were consistent with classic homocystinuria. The 14-month-old boy had normal newborn screening. The 7-year-old boy initially had an abnormal newborn screen for homocystinuria but second tier test that consisted of total homocysteine was normal, so his newborn screen was reported as normal. With the advent of expanded newborn screening many treatable metabolic disorders are detected before affected infants and children become symptomatic. Methionine is the primary target in newborn screening for homocystinuria and total homocysteine is a secondary target. Screening is usually performed after 24–48 h of life in most states in the US and some states perform a second screen as a policy on all tested newborns or based on when the initial newborn screen was performed. This is done in hopes of detecting infants who may have been missed on their first screen. In the United Kingdom, NBS using dried blood spot is performed at 5 to 8 days after birth. It is universally known that methionine is a poor target and newborn screening laboratories have used different cutoffs for a positive screen. Reducing the methionine cutoff increases the sensitivity but not necessarily the specificity of the test and increasing the cutoff will miss babies who may have HCU whose levels may not be high enough to be detected at their age of ascertainment. It is not clear whether adjusting methionine level to decrease the false negative rates combined with total homocysteine as a second-tier test can be used effectively and feasibly to detect newborns with HCU. Between December 2005 and December 2020, 827,083 newborns were screened in Kentucky by MS/MS. Kentucky NBS program uses the postanalytical tools offered by the Collaborative Laboratory Integrated Reports (CLIR) project which considers gestational age and birthweight. One case of classical homocystinuria was detected and two were missed on first and second tier tests respectively. The newborn that had confirmed classical homocystinuria was one of twenty-three newborns that were referred for second tier test because of elevated methionine (cutoff is >60 µmol/L) and/or Met/Phe ratio (cutoff is >1.0); all 23 dried blood spots had elevated total homocysteine. One of the subjects of this case report had a normal methionine on initial screen and the other had a normal second-tier total homocysteine level. The performance of methionine and total homocysteine as screening analytes for homocystinuria suggest that it may be time for newborn screening programs to consider adopting next generation sequencing (NGS) platforms as alternate modality of metabolic newborn screening. Because of cost considerations, newborn screening programs may not want to adopt NGS, but the downstream healthcare cost incurred due to missed cases and the associated morbidity of affected persons far exceed costs to newborn screen programs. Since NGS is becoming more widely available and inexpensive, it may be feasible to change testing algorithms to use Newborn Metabolic NGS as the primary mode of testing on dry blood specimens with confirmation with biochemical testing. Some commercial laboratories have Newborn Screening Metabolic gene panel that includes all metabolic disorders on the most comprehensive newborn screening panel in addition to many other conditions that are not on the panel. A more targeted NGS panel can be designed that may not cost much and eventually help avoid the pitfalls associated with delayed diagnosis and cost of screening.

## 1. Introduction

The homocystinurias are a heterogeneous group of metabolic disorders due to defects in the methionine metabolic cycle [1]. The first of these to be discovered and by far the most famous and best known is homocystinuria (HCU), which may better be described as classical homocystinuria to distinguish it from the other defects in the cycle in which elevated homocysteine is a key feature [2]. The basic defect is a marked reduction in activity of the key enzyme for transsulfuration, cystathionine β-synthase (CBS), and is biochemically characterized by elevated methionine and total homocysteine as well as reductions in cystathionine, the immediate product of CBS, and cysteine [3]. Other defects in the methionine metabolic cycle that result in homocystinuria are remethylation defects in which the conversion of homocysteine back to methionine is impaired. These defects produce reduced rather than increased methionine as well as increased homocysteine and may also be associated with an increase in methylmalonic acid. With one exception, methylene tetrahydrofolate reductase deficiency, they are collectively known as the cobalamin defects because they are produced by defects in vitamin B_12_ metabolism [4,5].

Classical homocystinuria due to cystathionine β-synthase deficiency affects an estimated 1 in 200,000 newborns in the US, but the true prevalence may be much higher since current newborn screening protocols that utilize methionine and, in many screening programs, total homocysteine as a second-tier test, may fail to identify all newborns affected by the disorder [6,7]. Patients who are undiagnosed are at increased risk for intellectual disability, progressive myopia followed by vision loss from lens dislocation, skeletal changes with scoliosis, osteopenia and osteoporosis, fractures, as well as thromboembolism with strokes and venous thrombosis. Patients identified through newborn screening programs are treated with a combination of a very low methionine diet consisting of medical formulas and low protein diet as well as betaine, vitamin B_12_, vitamin B_6_, and folate [8]. It is estimated that about 50% of CBS patients respond to vitamin B_6_ supplementation but almost all infants identified by newborn screening (NBS) are B_6_ non-responsive, so it is likely that many pyridoxine (B_6_) responsive patients who are more severely affected are either undiagnosed or have been clinically diagnosed but late treated.

Massachusetts began NBS for HCU in 1968, the first state to do so, but it was not until 2009 that HCU became part of the Recommended Uniform Screening Panel (RUSP). Newborn screening by MS/MS has improved the timing of diagnosis and outcome for children with many metabolic diseases. Methionine is the primary screening target but because of missed cases often due to high cutoffs, a few states have lowered the cutoff for methionine and use total homocysteine as a second-tier test in newborns whose methionine was borderline or above the cutoff level on primary screening [9]. Even with these adjustments of methionine levels, some infants may be missed, and testing protocols should therefore be considered to identify these individuals who are missed because of problems associated with reference ranges or who may not have sufficiently elevated methionine and/or homocysteine on the initial screen. The second-tier test is performed on the same dried blood spot (DBS) as is the primary screen with no additional contact with the newborn [7]. Methionine cutoffs posted for 183 laboratories in the CLIR has 30 µmol/L as the first percentile, 54 µmol/L as median and 100 µmol/L as 99th percentile. On the other hand, methionine in dried blood specimen for classical homocystinuria for a sample of *N* = 82 ranged between 37 and 802 µmol/L with a median of 121 [7]. The fact that there is overlap of dried blood spot methionine levels of normal and classical homocystinuria newborns suggest that regardless of the cutoff used, some NBS programs are going to miss cases. Missed NBS for other metabolic disorders have been described [10]. NBS may fail to identify variant forms of maple syrup urine disease [11,12]. There is no consensus about what is considered an acceptable false positive rate among NBS programs but there is agreement among programs that higher false positive rates are better than false negatives. Presumptive positive results may result in early identification of a disorder and early initiation of treatment. However, this early possible diagnosis and treatment advantage may be offset by parental stress and parent-child bonding problems [13].

In this paper we describe two patients missed on newborn screening who were diagnosed with HCU after they presented with seizures and cerebral venous thrombosis in infancy and childhood, respectively. We discuss some of the known pitfalls associated with biochemical targets that current newborn screening programs utilize, including postanalytical tools to detect newborns with classical homocystinuria, and provide suggestions for adoption of other modalities of testing to minimize morbidity to patients and health care cost.

## 2. Patient 1

A 14-month-old boy presented to an outside hospital with loss of consciousness, peri-oral cyanosis and seizure-like activity. A computed tomography (CT) brain scan showed venous hyperdense enlarged cortical vein over the right parietal convexity extending into the superior sagittal sinus that was consistent with thrombosis. Upon transfer to a tertiary hospital, a brain MRI /MRV indicated cerebral venous thrombosis and abnormal T2 prolongation visualized diffusely throughout white matter of both cerebral hemispheres and involving the corpus callosum, anterior commissure and internal capsule, on T2-weighted images (Figure 1). An electroencephalogram (EEG) showed generalized slowing with no epileptiform activity. He was started on levetiracetam. Blood for hypercoagulable work-up showed elevated total homocysteine of 260.61 µmol/L (normal range was 6.6 to 14.8). Quantitative plasma amino acids showed elevated methionine (118 µmol/L; normal range is 14–50) with markedly elevated homocystine (45 µmol/L; normal range is 0–2) that was consistent with a diagnosis of cystathionine β-synthase deficiency. Homocystinuria gene panel testing showed a paternally inherited likely pathogenic variant (c.904G>A) and a maternally inherited pathogenic variant (c.667-14_667-7del(intronic)) in the CBS gene. Plasma very long chain fatty acids, lysosomal enzyme panel, plasma acylcarnitine profile, urine organic acids, serum methylmalonic acid, and ceroid neuronal lipofuscinosis enzymes, palmitoyl-protein thioesterase 1 (PPT1) and tripeptidyl peptidase 1 (TPP1) were normal. Factor V Leiden and prothrombin gene mutation, protein S and antithrombin levels and antiphospholipid antibody, β-2 glycoprotein and anti-cardiolipin antibody were normal. Protein C activity was decreased (67% with refence range of 80% to 160%), folic acid was low, serum vitamin B_12_ was low-normal (285 with reference range of 211 to 946 pg/mL), and serum carnitines were low. RBC folate was 29.2% (reference range was 30% to 42%). Lipoprotein lipase A level was normal (17 mg/dL with reference range of less than 29 mg/dL) and serum methylmalonic acid was 0.12 (normal is 0.0 to 0.4 µmol/L). After the diagnosis of homocystinuria was confirmed, he was placed on a low methionine diet, folic acid, vitamin B_12_, vitamin B_6_, betaine supplementation and enoxaparin. Serial brain MRI showed complete resolution of the cerebral venous thrombosis with improvement of the white matter changes in the brain, so enoxaparin was discontinued, and he was placed on daily 81 mg aspirin. The patient remained alert and stable throughout his hospitalization and did not show any neurological signs of any intracranial pathology after the initial seizure-like episode.

Pregnancy history revealed that he was born at 31–32 weeks’ gestation to a 21-year-old gravida 2 para 1 mother and a 20-year-old father by cesarean section due to maternal pre-eclampsia. Birth weight was 1820 g and length was 42 cm. Newborn screening tests obtained at 3 days of age were normal. Blood spot methionine was 26.93 µmol/L (abnormal is >60; Met/Phe ratio was 0.50 with abnormal value >1.0); C3 was 1.83 and C3/C2 ratio was 0.06). He stayed in the NICU for 24 days and then discharged. He did not have a follow-up NBS. He had normal growth and development. Developmentally, he was age-appropriate and at 14 months, he was cruising and had several words. On physical examination, he was alert and interactive. Weight was at the 84th centile, length at the 69th centile and head circumference was at the 95th percentile. He did not have any unusual physical features. He has required gastrostomy tube placement for medical formula and medication management due to compliance problems. His neurological examinations have been normal; however, he has some developmental speech delay, and he is receiving speech therapy. His total homocysteine ranged from 58 µmol/L to 249 µmol/L on therapy.

## 3. Patient 2

A 7-year-old boy with history of hypertension, familial tall stature and generalized overgrowth was admitted for persistent vomiting, mild leukocytosis of 14,700 and mycoplasma pneumonia. Initial brain MRI was reported as normal. He developed generalized tonic clonic seizures and left hemiparesis on day 3 of hospitalization. Bedside glucose and serum electrolytes were normal. Ammonia was mildly elevated at 58 µmol/L but repeat was less than 9; lipase was 1273 U/L (normal is 23–300 U/L). An emergency head CT scan was read as normal. Neurology was consulted and he was transferred to the pediatric intensive care unit for closer monitoring. A video EEG in the ICU was suggestive of a right > left posterior quadrant cerebral dysfunction. Dilated funduscopic examination showed edema of the optic disc bilaterally without obscuring overlying vessels. Lumbar puncture showed an opening pressure greater than 38 cm of water (normal is less than 25 cm water) with normal cell count. He was started on acetazolamide for pseudotumor cerebri. MRV showed extensive dural venous sinus occlusive disease with extensive superficial and deep collateral veins and review of the previous brain MRI showed a filling defect in the superior sagittal sinus consistent with dural sinus thrombosis (Figure 2) and papilledema. Hypercoagulable work-up showed mildly decreased antithrombin III activity (80%; reference range was 83% to 128%). Factor V Leiden, prothrombin gene mutation, lupus anticoagulant and protein S were normal and protein C activity was low (63% with reference range of 70% to 140%). Metabolic work-up showed plasma homocysteine 71.9 (reference range was 6.6 to 14.8 µmol/L); plasma amino acids showed methionine of 448 µmol/L (reference range was 14 to 50) and free homocystine was 7 (reference range was 0–2 µmol/L). Serum methylmalonic acid was 0.15 (normal is 0.0 to 0.4 µmol/L), serum folate was 14.4 (reference range is 7 to 34.1 ng/mL), serum vitamin B12 was 244 (reference is 213 to 816 pg/mL). Blueprint Genetics Laboratory Homocystinuria Core Panel Plus showed a compound heterozygote pathogenic variant c.325T>C (p.Cys109Arg) and a variant of uncertain significance c1604C>T (p.Thr535Ile) in the CBS gene.

Pregnancy history revealed that the patient was born at 37 weeks’ gestation. Birth weight was 3830 g. He stayed in the NICU for 3 days because of transient tachypnea. Newborn screening tests performed at 2 days of age showed elevated methionine 89.4 µmol/L (abnormal is >60; Met/Phe ratio was 2.0 with abnormal value >1.0). Second tier test reported normal total homocysteine level. The total homocysteine on the DBS was 14.9 µmol/L. The reference laboratory’s reference range was less than 15 µmol/L.

Developmentally, he walked at 15 to 17 months. He required physical and occupational therapies for fine motor delays. Cognitively he was age appropriate and performing at grade level in school. Physical examination showed a tall boy with weight and height above the 99th percentile. His BMI was above the 99th percentile. He did not have Marfanoid features. He presented to his primary care physician 8 months after his diagnosis of HCU with back pain that was worse with bending forward and walking. He was seen by a spine surgeon and a spine MRI showed multiple chronic compression fractures from T6 to L4 vertebrae with osteoporosis. No scoliosis was reported. Bone densitometry scan showed that lumbar mineralization was 82% of mean for age and gender which is consistent with osteopenia. Left hip mineralization was 96% that is normal but there was osteopenia of the femoral head. He has also had right ankle avulsion injury versus old fracture and left distal fibula fracture. He was referred to pediatric endocrinology who after consultation with genetics prescribed pamidronate therapy for him. His treatment of HCU with Medical foods and medication continue to be a challenge at his age and his total homocysteine has been mostly above 100 µmol/L with a range of 66 µmol/L to 217.4 µmol/L on betaine therapy and low methionine medical foods.

## 4. Discussion

With the advent of expanded newborn screening (ENBS) using MS/MS, many treatable metabolic disorders are being detected before affected infants and children become symptomatic. Screening is usually performed within 48 h of life in most states in the US and some states perform a second screen as a policy on all tested newborns or based on when the initial newborn screen was performed [9]. This is done in hopes of detecting infants who may have been missed on their first screen [8]. In the United Kingdom, NBS is done 5 to 8 days after birth, and this may give newborns the opportunity to establish full feedings that may help improve detection of some analytes [14,15,16].

The primary newborn screening target for homocystinuria is methionine. It is well established that with methionine as the primary target, infants are missed on the screen, but it is not clear how many infants are not detected. The cutoff values for methionine vary among NBS programs and this may reflect diversity of populations screened and timing of blood spot collection. Laboratories have adjusted their cutoffs downwards and may have helped minimize the number of cases that are missed but has not been eliminated completely. The Mayo Clinic Region 4 program has the CLIR algorithms that utilize total homocysteine and other analytes as second tier test for classical homocystinuria and homocystinurias due to cobalamin defects [17]. In the UK, the cutoff of methionine is 45 µmol/L and the total homocysteine cut-off is 15 µmol/L [14].

Missed NBS for other metabolic disorders such as glutaric acidemia type 1 and very long chain acyl-coA dehydrogenase deficiency have been described [10]. It has been reported that NBS may fail to identify variant forms maple syrup urine disease [11,12]. Methionine cutoffs posted for 183 laboratories in the CLIR has 30 µmol/L as the first percentile, 54 µmol/L as median and 100 µmol/L as 99th percentile. On the other hand, methionine in dried blood specimen for classical homocystinuria for a sample of *N* = 82 ranged between 37 and 802 µmol/L with a median of 121 [7,17,18]. The fact that there is overlap of dried blood spot methionine levels of normal and classical homocystinuria newborns suggest that regardless of the cutoff used, some NBS programs are going to miss cases. Implementation of MS/MS-based second tier tests have led to reductions in presumptive positive rates in NBS [7]. Presumptive positive results may result in early identification of a disorder and early initiation of treatment. However, this early possible diagnosis and treatment advantage may be offset by parent stress and parent-child bonding problems [13].

Potential causes of false negatives include blood transfusion, feeding not well established, delays in transit leading to sample deterioration and physiological reasons; pyridoxine responsive infants are less likely to be detected. Some of the potential causes of presumptive positives include total parenteral nutrition and liver disease. Infants with methionine adenosyltransferase deficiency may have elevated methionine and raised total homocysteine could be due to rare inborn errors of metabolism such methylenetetrahydrofolate reductase deficiency and defects of vitamin B 12 metabolism [14].

To improve performance of NBS for the homocystinurias, a four-step strategy has been recommended: revision of cutoffs based on local median values, combination of relevant markers such as Met/Phe, use of the center-adjusted postanalytical tools offered by CLIR and implementation of total homocysteine [6]. Beginning in May 2021, Kentucky NBS program lowered methionine cutoffs to 50 µmol/L and all newborns automatically get total homocysteine level above that cutoff. Our reference laboratory, Mayo Clinic, has also lowered their total homocysteine cutoff to 9 µmol/L.

NGS platforms have been explored as possible primary screening modality [11,12,19]. If implemented, this may greatly reduce false positive results and decrease unnecessary stress on families due to recalls. It can be useful in distinguishing affected individuals from carriers.

We present 2 boys who were missed on NBS 2 to 3 days after birth and presented with cerebral venous thrombosis at a later age and in whom metabolic work-up revealed cystathionine β-synthase deficiency. Patient 1 had a normal methionine. He was born premature, and his birth weight was less than 2000 g so he may have benefited from repeat NBS. Patient 2 had an initial abnormal newborn screen but the second-tier total homocysteine test on dry blood spot was normal, so he was considered to have normal screen for homocystinuria. Between December 2005 and December 2020, 827,083 newborns were screened in Kentucky by MS/MS. Kentucky NBS program uses the postanalytical tools offered by the CLIR project which considers gestational age and birthweight. With an estimated prevalence of 1 in 200,000, at least four cases of HCU were expected to be detected using the current NBS methodologies. One case of classical homocystinuria was detected and two were missed on first and second tier tests, respectively.

It is unknown how many infants are missed on initial single newborn screen. It may therefore be reasonable for newborn screening programs to use combination analytes such as methionine and total homocysteine, methionine/phenylalanine ratio or adopt a mandatory confirmatory metabolic testing of conditions with significant risk of false negative results to avoid the pitfalls associated with possible missed cases on initial screen.

As NGS becomes more widely available and inexpensive, it may be feasible to change testing algorithms that employ biochemical methods only to adoption of Newborn Metabolic NGS as the primary mode of testing on DBS with confirmation with biochemical testing as is currently done. All variants of unknown significance will have biochemical testing to confirm the possible pathogenicity of the variants identified. NGS on DBS have been piloted and found to detect causal pathogenic variants in patients with inherited metabolic diseases [19]. NGS panels targeting the entire coding regions of genes relevant to newborn screening, rapid turnaround time and cost have been determined to be feasible [16,19,20]. Some commercial laboratories have Newborn Screening Metabolic gene panel on their test menu that includes all metabolic disorders on the most comprehensive screening panel in addition to many other metabolic conditions that are not on the panel. Rethinking delivery of NBS and employing molecular genetic diagnostic approaches may help avoid some of the pitfalls associated with missed screens and the health care cost and attendant morbidities associated with missed biochemical screening.

## Figures and Tables

**Figure 1 IJNS-07-00048-f001:**
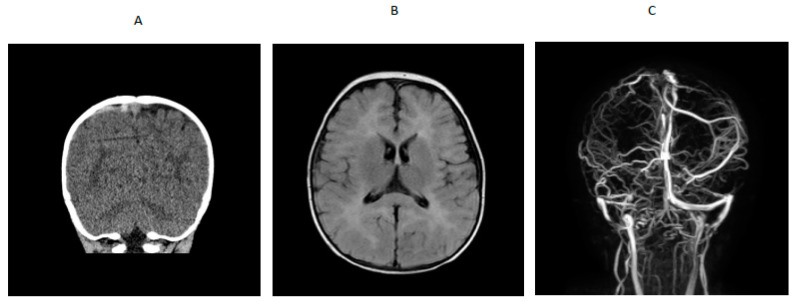
(**A**) CT scan showing venous hyperdense cortical vein over right parietal convexity. (**B**) MRI showing extensive white matter changes. (**C**) MRV showing thrombosis of right parietal cortical vein extending into the superior sagittal sinus.

**Figure 2 IJNS-07-00048-f002:**
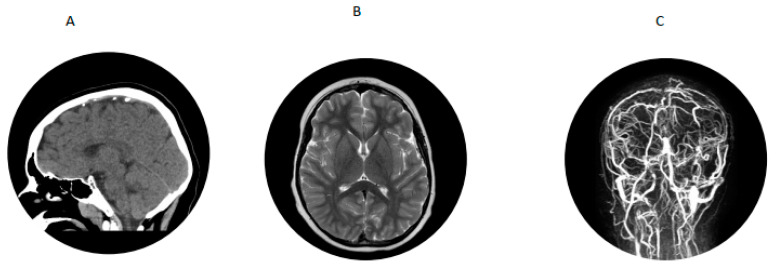
For patient 2. (**A**) CT scan showing thrombosis of superior sagittal sinus. (**B**) Brain MRI showed filling defect in superior sagittal sinus. (**C**) MRV showing dural venous thrombosis with extensive superficial and deep collateral veins.

## Data Availability

Exclude because study did not report any data.

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
