# Peer review of "Diagnosis of Classic Homocystinuria in Two Boys Presenting with Acute Cerebral Venous Thrombosis and Neurologic Dysfunction after Normal Newborn Screening"

_2409-515X, 2021, doi:10.3390/ijns7030048_

Round 1
Reviewer 1 Report
This case report details two false negative cases of HCU. It is well recognised that using methionine as a first line screening test for HCU will miss cases with pyridoxine non-responsive HCU. Whilst these cases are interesting the report lacks essential information to make it a useful addition to the literature.
Case 1 – No details of the actual methionine screening result was included? Was the methionine result close to the screening action value? What was the performance of the assay (i.e. was there a negative bias for the assay)? Was the screening sample retrieved and re-analysed for total homocysteine?
Case 2 – It is stated that there was an abnormal screening result (methionine 89.4µmol/L) and that the bloodspot total homocysteine was normal… what was the homocysteine concentration and what action value/cut off is used? Was the result near to the cut-off? What is the performance of the assay?
The methionine result of 89.4µmol/L and the met/phe ratio of 2.0 in case 2 – how does this value compare with other known true positive cases from this programme? Has the screening sample been retrieved and re-analysed for total homocysteine to confirm the result?
Discussion - During the period of screening in this region/laboratory how many infants have been screened? What proportion were false positive and false negatives? Does the methionine screening cut-off value need to be reconsidered? How does this compare with those countries that screen at day 5 of life?
Author Response
Responses to Reviewer 1 comments
We have addressed
Case 1.
Methionine was 26.93 µmol /L (normal reference is <60), Methionine/phenylalanine ratio was 0.50 (normal reference is <1.00). No second tier was performed because the initial screen was normal. C3 was 1.83; C3/C2 ratio was 0.06 (cutoff are 4.8, 0.2 and 3.0 respectively) based on CLIR post-analytic tools.
Case 2.
Methionine was 89.4 µmol/L /L (normal is <60) and Methionine/Phenylalanine ratio was 2 (normal is <1.0). The cutoff values for Methionine, MET/PHE ratio, C3, C3/C2 ratio and C3/C16 ratio used in the newborn screening laboratory are 60, 1.0, 4.8, 0.2 and 3.0 respectively, based on CLIR post analytic tools and there is no negative bias for methionine. Any of the analytes with the primary screen above the cutoff above is sent to a reference laboratory for second tier total homocysteine, cobalamin, methylmalonic acidemia tests. The blood spot was reported as negative for total homocysteine by the reference laboratory (normal reference for total homocysteine was <15). The total homocysteine of this case was 14.9 µmol/L . The reference laboratory has recently changed its total homocysteine cutoff level to 9.0 µmol /l. We recently had a case of confirmed classic homocystinuria. She was born at term and weighed 3440 g and blood spot was collected after 24 hours of life. Methionine was 77.4 µmol /L (normal is <60), MET/PHE ratio was 1.50 (normal is <1.0). Second tier test showed total homocysteine of 51.3 µmol/L (normal is <15), methylmalonic acid was 0.9 µmol /L (normal is <5) and Methylcitric acid was 0.1 /L (normal is <1.0). Confirmatory testing was positive for classical homocystinuria.
Residual dried blood specimens are not used for research as defined by the common Rule of The Newborn Screening Saves Lives Reauthorization Act of 2014. Accordingly, NBS lab policy states that upon completion of laboratory tests, samples be stored for a period of 2 months. Presumptive positive cases are stored indefinitely in a secure location and may be used for internal purposes such as quality control, method verification and method validation studies. Since case 2 was considered screen-negative, the specimen was not stored.
Discussion: This has been revised with more references.
Since expanded NBS started, 827,083 newborns were screened between December 2005 and December 2020, giving an average of 51,000 a year. There has been one confirmed classical homocystinuria after an initial positive screen and a second-tier test (2TT). Twenty-three specimens with elevated total homocysteine from 2TT were referred for final thorough diagnostic workup but only one had confirmed HCU diagnosis. Beginning on May 10, 2021, the Kentucky NBS program has lowered methionine cutoff to 50 µmol/L and then referred for total homocysteine as 2TT in reference laboratory as usual practice. Patients who had elevated total homocysteine are then referred to specialty geneticist for full diagnostic workup to clarify final diagnosis. We do not have any good way to compare the practice in our region where newborns are screened after 24 hours of birth with other countries that screen after 5 days of life. The UK screens at 5 to 8 days after birth but their healthcare system is different from that of the US. Since the inception of expanded NBS we have had 23 elevated total homocysteine after second-tier test and only one patient had confirmed classical homocystinuria. Patient 2 was referred because of abnormal Methionine but total homocysteine was normal that was deemed screen negative. Case 1 had normal methionine so did not qualify for second-tier testing. However, because he was born premature with a birthweight less than 2500 grams, repeat testing should have been performed. For a state with an average of 51000 screening annually, we expect to see one case every 4 years assuming HCU prevalence of 1 in 200,000. It is possible that there are B6-responsive milder cases that are missed or because of maternal prenatal vitamin use.
Reviewer 2 Report
This article shows two cases with HCU detected after onset despite undergoing NBS. However, it is well known that a use of Met and tHcy levels to detect individuals with HCU isn't sufficient. Therefore, these cases are not so rare. Furthermore, the discussion part is poor.
<Major comments>
These cases should be more discussed about causes of false positive. For example, how were the nutrition condition including the volume of breast feeding or milk at date of sampling dried blood spots? Were they Vit B6 responders? Please detail several factors of false negative in NBS.
Additionally, please note what the authors should do to avoid the false negative. For instance, if Met/tHcy and Met/Phe ratios were used for NBS, can these patients be detected? Or, any markers including second tier could not detect these patients at NBS? Moreover, did the author foster the momentum of lowering the cutoff levels in your institution through these false-negative cases?
<Minor comments>
Some abbreviations are not spelled out. For example, CT, EEG, MRI, MRV, and STAT.
Because “Keppra” is a commodity name, please use levetiracetam.
There are no units of ammonia, lipase, and opening pressure of lumbar puncture in Patient 2.
Please show the clinical course, such as neurological outcomes and Met levels in serum, after treatment in patient 2.
Please more describe clinical information, such as typical symptoms of HCU including arachnodactylia and scoliosis, past history of bone fracture, and limb paralysis. Furthermore, images of brain MRI and CT in acute phase may be important.
Author Response
Response to Reviewer 2 comments.
Major comments
Met/Phe ratio is used but Met/Hcy ratio is not used by our screening program. It appears in one case that was missed, the Met/Phe ratio was above the reference range. It is not clear if the Met/Hcy was part of the tool used to call screen-positive, that would have made a difference in detecting these missed cases.
Case 1 was born premature (31-32 weeks’ gestation) and his blood spot was collected at 3 days of age. He was in the neonatal intensive care unit for 24 days. It is not clear whether at the time of his newborn screening, he was on full feeding or not. If he was not on full feeds at the time of the screen, then he may not have had enough protein challenge to cause analytes to build up and this may have been one of the factors that caused the false negative result. This patient should have had a second newborn screen for the simple reason that he was born premature.
Case 2 was born at term. Newborn screening was performed after 48 hours of birth. He had elevated methionine and Met/Phe ratio but his total homocysteine was just below the cutoff so it was reported as normal. We cannot explain why total homocysteine was normal in the dried blood spot and the question is whether in cases like this total homocysteine should be obtained in whole blood rather than dried blood spot. Again patient was not vitamin B6 responsive so maternal prenatal vitamin use will not have made a difference in his outcome.
Minor comments:
All abbreviations used have been spelt out. Keppra has been changed to Levetiracetam.
Units for ammonia and lipase have been provided and the opening pressure of the lumbar puncture has been provided.
The clinical course and total homocysteine levels after treatment have been added to the information on patient 2. Total methionine levels were high because of betaine therapy so we did not think they were useful information. Further medical history after HCU diagnosis has been added and images of brain MRI and CT in acute stages included under the figures.
Reviewer 3 Report
In case 1, the newborn screening result is normal but have some newborn screening results in case 2. If you find the newborn screening data of case 1 or test the retrieved DBS samples, it will be valuable.
Author Response
Response to Reviewer 3 comments
Case 1: Methionine was 26.93 mmol/L (normal reference is <60), Methionine/phenylalanine ratio was 0.50 (normal reference is <1.00). No second tier was performed because the screen was normal.
Round 2
Reviewer 1 Report
the authors have addressed the points raised.
Author Response
Reviewer has accepted revisions to points raised
Reviewer 2 Report
The authors almost completely answered my questions and suggestions. However, new minor problems occur in the part that I did not request to correct. For example, "next generation sequencing" was used after abbreviating NGS in the abstract, and "presumptive positive results" on line 15 of 3rd page and 6th page may be appropriate (not false positive). As well, the use of abbreviation is sometimes inappropriate. "computerized tomography" and "Collaborative Laboratory Integrated Reports" were used after abbreviating. "HCU" was spelled out twice. It is also inappropriate that a specific brand name such as "Invitae" is written in an abstract.
Author Response
Reviewer 2 accepted all the major points addressed in first revision. Reviewer raised minor points which we have addressed in second revision and tracked.
- Next generation sequencing abbreviation was defined so going forward NGS was used.
- False positive was changed to "presumptive positive results" in all places applicable (page 3 line 15, page 6)
- Computerized tomography abbreviation was defined so CT was used going forward.
- HCU had been defined so did not use classic homocystinuria with HCU in parenthesis together.
- Collaborative Laboratory Integrated Reports had been defined so CLIR abbreviation was used going forward.
- I removed the brand name "Invitae laboratories" and replaced with "Some commercial laboratories"
I hope these address all the points raised by reviewer 2. Thanks.